# Detection of SARS-COV-2 Proteins Using an ELISA Test

**DOI:** 10.3390/diagnostics11040698

**Published:** 2021-04-14

**Authors:** Marina Di Domenico, Alfredo De Rosa, Mariarosaria Boccellino

**Affiliations:** 1Department of Precision Medicine, University of Campania “Luigi Vanvitelli”, 81100 Naples, Italy; marina.didomenico@unicampania.it; 2Department of Biology, College of Science and Technology, Temple University, Philadelphia, PA 19122, USA; 3Multidisciplinary Medical-Surgical Department, Odontostomatology Section, University of Campania “Luigi Vanvitelli”, 81100 Naples, Italy; alfredo.derosa@unicampania.it

**Keywords:** COVID-19, SARS-CoV-2, ELISA, cytosalivary sampling

## Abstract

The coronavirus disease 2019 (COVID-19) global pandemic created an unprecedented public health emergency. Early recognition of an infected person and disruption of the transmission pathway are the keys to controlling this major public health threat around the world. The scientifically reliable screening method is an RT-PCR test that is performed on an ororhinopharyngeal swab in the laboratory. In the current severe SARS-CoV-2 pandemic, it is necessary to identify devices for rapid diagnosis to reduce the spread of the disease. The aim of this study was to provide a qualitative, rapid, sensitive, and specific method for a diagnosis of SARS-CoV-2 infection based on the recognition of specific antigens of the SARS-CoV-2 virus. The device was built by assembling commercially available and custom-made semi-finished products. The method was performed in environments outside the laboratory, i.e., “patient side,” with an immediate chemocolorimetric response or with a digital reader using an ELISA method.

## 1. Introduction

In Wuhan, China, a group of pneumonia patients of unknown cause was identified and linked to a seafood wholesale market in December 2019. A new type of coronavirus belonging to the genus β was identified by sequencing samples from subjects with pneumonia. The newly isolated coronavirus belonging to the sarbecovirus subgenus of the Orthocoronavirinae subfamily was named 2019-nCov. Six species of coronavirus are already known to cause disease in humans: four viruses 229E, OC43, NL63, and HKU1 are prevalent and cause common cold symptoms in immunocompetent individuals; two other strains, namely, severe acute respiratory syndrome coronavirus (SARS-CoV) and Middle Eastern respiratory syndrome coronavirus (MERS-CoV), are zoonotic in origin and have been linked to sometimes fatal diseases [1]. Thanks to its close relationship to the virus responsible for the SARS epidemic (SARS-CoV) in 2003, the novel coronavirus was soon called severe acute respiratory syndrome coronavirus 2 (SARS-CoV-2). Coronavirus disease 2019 (COVID-19) is the name of the disease caused by this new virus. Although there is an approximately 80% sequence similarity with SARS-CoV, the contagiousness of SARS-CoV-2 is much higher, as demonstrated by the worldwide spread of the infection, with over 40,000,000 officially diagnosed cases and 1,128,000 deaths by the end of October 2020 [2]. On 11 March 2020, the World Health Organization (WHO) declared the spread of SARS-CoV-2 a pandemic [3]. In the global coronavirus epidemic, many SARS-CoV-2 infections are asymptomatic or have only mild symptoms but can still transmit the virus to others. Indeed, asymptomatic infections refer to the positive detection of SARS-CoV-2 nucleic acid in patient samples using reverse transcription–polymerase chain reaction (RT-PCR), but they have neither clinical signs nor typical symptoms and no abnormalities are apparent in images, such as lung computed tomography (CT) [4]. The infectivity of asymptomatic infections is the same as that of symptomatic infections. There are several studies on the incidence of asymptomatic infections that explain the potential of the epidemiological transmission of COVID-19, as well as its universality. A study conducted on 565 Japanese citizens that were evacuated from Wuhan in January 2020 showed that the incidence of asymptomatic infections was 30.8% [5]. On the “Diamond Princess” cruise ship, which was isolated in Japanese waters in February 2020 due to SARS-CoV-2 infections, an incidence of asymptomatic infections of 51.7% was found [6]. There are several difficulties regarding screening for asymptomatic infections; however, early recognition of an infected person and disruption of the transmission pathway are the keys to controlling this major public health threat around the world. The detection of viral RNA using RT-PCR in respiratory specimens was recognized as the gold standard for the diagnosis of a SARS-CoV-2 infection. However, the main limitation of this technique is that it takes several hours to generate results; in addition, specialized instruments and expertise are required. All this reduces the possibility of making rapid diagnoses directly in the field and its use in a mass screening program. In the current severe SARS-CoV-2 pandemic, it is necessary to identify devices for making rapid diagnoses to reduce the spread of the disease. The aim of this study was to provide a qualitative, rapid, sensitive, and specific method for the diagnosis of a SARS-CoV-2 infection based on the recognition of specific antigens of SARS-CoV-2 that can also be performed in environments outside the laboratory, i.e., “patient side,” with an immediate chemocolorimetric response or with a digital reader.

## 2. Materials and Methods

### 2.1. Antibodies

SARS-CoV/SARS-CoV-2 (COVID-19) spike antibody (1A9]; SARS-CoV/SARS-CoV-2 (COVID-19) nucleocapsid antibody (6H3) Cat; Mouse IgG (Fc fragment) antibody, F(ab’) 2 fragment, pre-adsorbed (AP); Rabbit anti-SARS Virus Spike Protein; Rabbit anti-SARS Virus Nucleocapsid Protein; 2019-nCoV Spike Protein (S1+S2 ECD, His tag); 2019-nCoV Nucleocapsid Protein (His tag); Actin antibody were purchased from GeneTex (Irvine, CA, USA). PVDF strips were purchased from ThermoFisher Scientific (Waltham, MA, USA).

### 2.2. PVDF Strip Preparation Protocol 

The steps for preparing the PVDF strips armed with the primary antibodies of interest were as follows: Hydration of the PVDF strip with methanol for 5 min, followed by two washes with PBS for 5 min. Incubation with protein A 10 µg/mL for 1 h in PBS followed by two washes with PBS. After washing, the strips were blocked with 3% BSA solution in PBS for 1 h and washed three times with PBS for 5 min. The PVDF strips were then separately incubated with a solution containing rabbit antibody anti-SARS Virus Spike Protein or anti-SARS Virus Nucleocapsid Protein or anti-actin antibodies at a concentration of 3 µg/mL overnight a 4 °C via shaking. The strips were then washed three times with PBS for 5 min, followed by two washes with 0.2 M PBS TEA. The strips were further incubated with 25 mM DMP in 0.2 M TEA HCl, pH 8.2, followed by a solution containing 0.2 M TEA + 20 mM ethanolamine) and two washes with PBS for 5 min. The strips were then stored in 0.02% NaN_3_ in PBS. The strips were cut with a width of approximately 0.4 cm in order to fit on the device support to perform the ELISA analysis (Figure 1).

### 2.3. Device Design

The device used dried or lyophilized antibodies, which were stable at room temperature; at the time of use, they were solubilized in a suitable buffer solution. The device was built by assembling commercially available and custom-made semi-finished products. It included an instrument such as cytobrush and/or nasal swab and/or throat swab (ATS Sanitary Supplies, Rome, Italy), which was useful for taking the biological sample at the level of the ororinopharyngeal mucosa; this was necessary to analyze it and verify the presence of viral biomarkers by using the ELISA method. The product had an extremely simple method of use.

### 2.4. Description of the Device

The kit box was organized in rows of nine wells/stations containing buffers, lysis systems, and detection systems, as well as dried and/or lyophilized antibodies, all of which were stable at room temperature. Furthermore, the kit was equipped with tools for the collection of biological material and support for the PVDF strip (ThermoFisher Scientific, Waltham, MA, USA), which was armed with the primary antibodies of interest. 

### 2.5. Test Procedure

The reagents found in the wells/station in a dried and/or lyophilized state were dissolved into an appropriate buffer when opening the kit.The cells from the ororhinopharyngeal mucosa and potentially infected sputum were collected with specific instruments.The buffer was immersed with the biological sample for 8 min in the well/station 0 containing the lysis buffer.The lysate was transferred into well/station 1.The package of the support was opened to whichever PVDF membrane was adhered to, which was then immersed in well/station 1 for 8 min for the recognition of viral antigens by the primary antibodies immobilized on the PVDF strip. In this station, immune complexes were formed on the strip if there was the presence of viral proteins in the sample taken.The PVDF strip with the immobilized immune complexes was immersed for 5 min in well/station 2, where there were primary monoclonal antibodies in the solution that bound specifically to the immunocomplex.In wells/stations 3 and 4, the PVDF support holding the strip was washed in wells containing the T-PBS buffer to eliminate the proteins that were not attached to the immunocomplex.The support was immersed with the PVDF strip for 5 min in well/station 5 containing the secondary antibodies conjugated to an enzymatic detection system (alkaline phosphatase).In wells/stations 6 and 7, washings of the support with the PVDF strip took place in wells containing the T-PBS buffer to eliminate any excess secondary antibody.The support with the PVDF strip was immersed for 4 min in well/station 8 containing the substrate (BCIP/NBT, stable at room temperature) that was necessary for the colorimetric reaction.The last phase consisted of the interpretation of the result.The colorimetric assay could be carried out in a temperature range between 20 and 30 °C. In fact, the alkaline phosphatase or peroxidase linked to secondary antibody possessed the maximum enzymatic activity in this temperature range, thus allowing for optimal signal amplification.

## 3. Results

This method was based on the direct assessment of SARS-CoV-2 virus particles through the determination of the spike and nucleocapsid proteins using an ELISA test. It allowed for qualitatively identifying the presence of antigens characterizing coronavirus infection by means of a cytosalivary sample. In this test, a series of antibodies against the specific protein was adsorbed onto a membrane. The membrane was then immersed in a cell lysate. If the desired marker proteins were present in the lysate in a concentration above the minimum signal threshold (limit of detection, LOD), these were captured on the membrane, which was subsequently immersed in a solution containing secondary antibodies that were conjugated with alkaline phosphatase, forming a sandwich that was detectable with a chemocolorimetric technique. The membrane was then immersed in the chemocolorimetric detection station where, in the presence of the complex, a pink/purple band was revealed. 

In order to evaluate the test specificity and sensitivity, we performed in vitro investigations with the aim to validate the method. For this reason, as a first step, we determined the best amount of primary polyclonal antibodies that should be bound to the PVDF membrane. We chose a concentration of 3 µg/mL in PBS for both polyclonal antibodies against spike and nucleocapsid proteins for incubation overnight at 4 °C. These antibodies guaranteed specificity toward the proteins under examination because they do not cross-react with SARS-CoV or MERS-CoV nucleocapsid and spike proteins based on internal testing performed by the manufacturer. The same concentration and time of incubation were used for the rabbit antibody anti-actin that was used as a control. Each antibody was bound on a different PVDF strip. The support with the strips was then immersed in a 400 µL solution containing a cell lysate, to which the proteins under examination with a concentration ranging from 15 µg/µL to 50 ng/µL were added (Figure 2); in this case, a very intense pink/purple color was obtained at the end of the ELISA test.

Subsequently, other tests that were carried out with lower protein concentrations from 500 pg/µL up to 5 pg/µL still showed a positive result, although with a pale pink color (Figure 3). Subsequent tests performed with protein concentrations lower than 5 pg/µL demonstrated the ability of the method to detect concentrations as low as 2 pg/µL; however, as part of the practical use of the device, in order to have an adequately detectable signal, we recommend 5 pg/µL as the sensitivity limit.

After we assessed the primary antibody concentration, we tested the second series of primary antibodies directed against the same viral antigens. For this, we plunged each strip into a solution containing a primary monoclonal antibody for each antigen at concentrations ranging from 1.5 µg/mL to 0.5 µg/mL in order to develop the sandwich. The in vitro results showed that even at a concentration as low as 0.5 µg/mL, the detection of viral proteins was effective. Finally, we also evaluated the optimal concentration of the secondary antibody; for this purpose, various tests were conducted using concentrations ranging from 1.5 µg/mL down to 0.2 µg/mL. The results showed good color detection down to the lowest concentration used (data not shown).

## 4. Discussion

The COVID-19 global pandemic created an unprecedented public health emergency. Currently, the best defense against this virus is attributed to social distancing and quarantine of the infected or positive asymptomatic population. Therefore, rapid screening becomes of primary importance to direct physicians toward the need for a treatment plan, and above all, to order the isolation of both symptomatic and asymptomatic subjects from healthy subjects as soon as possible, or similarly, avoid unnecessary and harmful quarantines of patients with symptoms of COVID-19 but lacking the SARS-CoV-2 virus due to the socio-economic harms that this can cause. At the moment, the scientifically reliable screening method is the RT-PCR test performed on an ororhinopharyngeal swab in the laboratory, which involves a significant economic expenditure and long lead times. These delays significantly reduce the effectiveness of the quarantine protocols and increase the risk of exposure to people close to the subject waiting for the result. The high cost also limits the use of large-scale testing. Another method of screening involves rapid tests, often involving the lateral flow technique, which uses blood or serum to look for specific antibodies, such as IgM and IgG, that are developed by patients who have contracted the SARS-CoV-2 virus. However, scientific studies have shown a low efficacy of these tests.

Since the beginning of the SARS-COV-2 pandemic, several studies have been published that investigated the infection by determining the quality and quantity of the antibodies produced. As far as we know, this is the first paper reporting a method to discover the infection via the direct determination of SARS-COV-2 proteins. 

The method is based on the ELISA technique with a chemocolorimetric result on a membrane with adsorbed antibodies for the specific virus antigens found in the oral sample. In particular, the kit we proposed allows for the determination of two SARS-CoV-2 proteins, the S-protein (spike protein) and the N-protein (nucleocapsid) by means of the dual use of primary antibodies to unequivocally guarantee the specificity and sensitivity of the signal. Specifically, a series of primary anti-SARS Virus COV-2 Spike Protein and anti-SARS Virus COV-2 Nucleocapsid Protein antibodies is loaded on a PVDF membrane, while another series of primary antibodies directed against the same proteins but from different animals are used in test tubes. Subsequently, secondary antibodies are used, which are conjugated with an enzymatic signal amplification system, such as alkaline phosphatase or peroxidase. Finally, the PVDF membrane is immersed in a chemocolorimetric detection station containing the substrate (BCIP/NBT) that is necessary for the colorimetric reaction; if hybridomas are present, a pink/purple stripe is revealed. The method can be assembled as a kit, which makes it easily transportable and usable in the field, i.e., “patient side,” as it does not need any particular equipment. The kit uses reagents that are stable at room temperature and it is rapid, sensitive, specific, and noninvasive. The test allows for qualitatively identifying the presence of antigens characterizing SARS-CoV-2 infection by means of a cytosalivary sample. It is useful for assisting a physician’s decision to proceed with further qualitative and quantitative tests for diagnosis. The search for antigens makes it possible to identify the incubation phase or the early stages of infection, where the antibody response is still low and the characterizing symptoms are not yet evident.

## 5. Conclusions

The present invention was configured as an effective diagnostic tool for detecting SARS-CoV-2 infections. Our device allows for performing a test that directly evaluates the presence of the virus in the biological sample taken by detecting its specific proteins (nucleocapsid and spike). It is quick; in fact, it takes about 30 min. It offers the advantage that the result of the antigen–antibody reaction can be directly visible to the naked eye without the need to be carried out in a laboratory, without analytical instrumentation, and above all, without the need to transport the samples. The high sensitivity of the method guarantees an early diagnosis, even in those subjects with borderline SARS-CoV-2 positivity. Its signal cut-off point makes it unique among diagnostic systems; in fact, our test has a particularly low sensitivity limit. Another strength of our test that is not present in other systems is that of simultaneously detecting two specific viral antigens, which certainly increases the specificity. Furthermore, the use of a cell lysis buffer in the first step of our kit allows for obtaining a greater viral load in solution, which accentuates the sensitivity of our method. In addition, it is a noninvasive test, for which there is greater acceptability by the subjects; the ease of the sample collection and the simplicity of the execution means that there is less need for operator training. Finally, the low cost allows for greater repetition. However, a limit is that a sample can be negative if the concentration of the antigens is lower than the detection limit of the test, which can happen if the sample is taken too early relative to the hypothetical moment of exposure. For this reason, the negativity of the sample in the face of strong suspicion of SARS-CoV-2 infection should be confirmed using a molecular test. Overall, the response speed of our test and the high sensitivity make it possible to monitor the incidence and progress of the disease, as well as evaluate its severity over time; moreover, it allows for operating a mass screening that can quantify the epidemic state of community environments, such as schools, airports, stations, and ports, preventing clusters or outbreaks.

The method has been described in a patent application titled “Methode de depistage in vitro cote patient pour le diagnostic rapide du Sars-CoV2,” inventors: Mariarosaria Boccellino, Marina Di Domenico, Alfredo De Rosa, Riccardo Moffa (application no. 2698/April 2020, Department des Finances et de l’Economie Principaute de Monaco). In vivo screening will be required for validating the in vitro method.

## Figures and Tables

**Figure 1 diagnostics-11-00698-f001:**
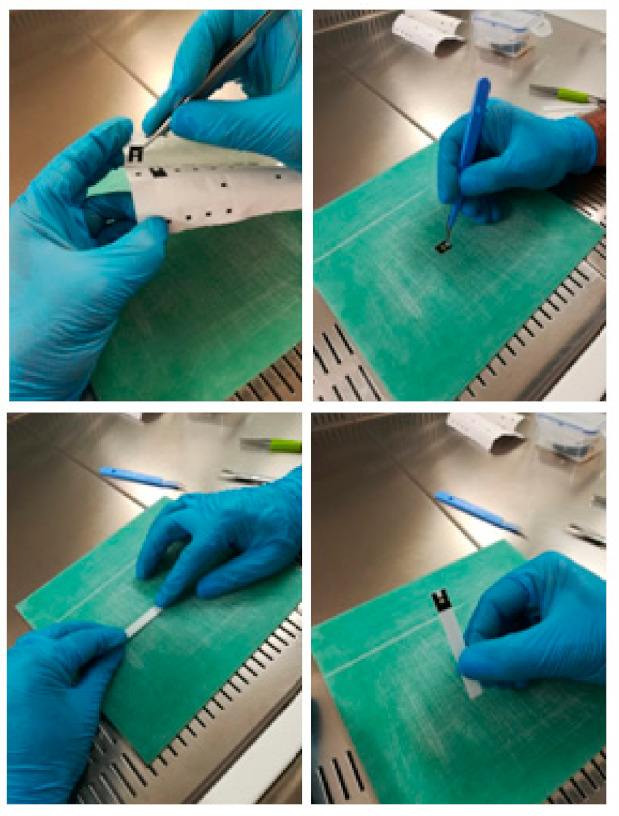
Stages of the preparation of a PVDF strip.

**Figure 2 diagnostics-11-00698-f002:**
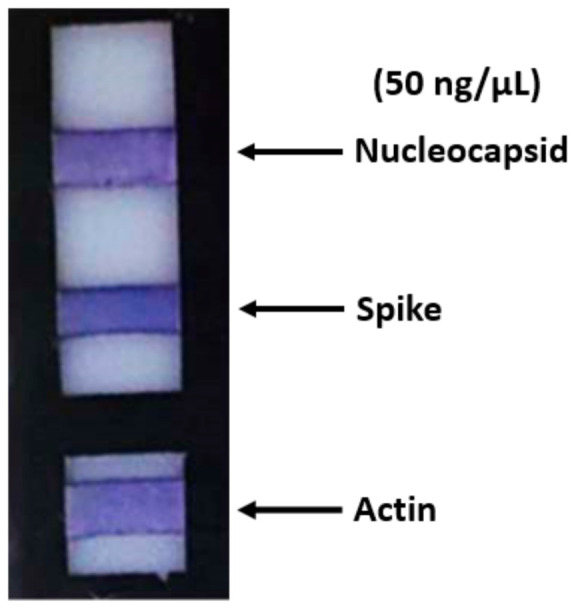
The ELISA assay. The support with the strips was immersed in a solution containing a cell lysate (range: 15 µg/µL–50 ng/µL). A very intense pink/purple colorimetric effect was obtained.

**Figure 3 diagnostics-11-00698-f003:**
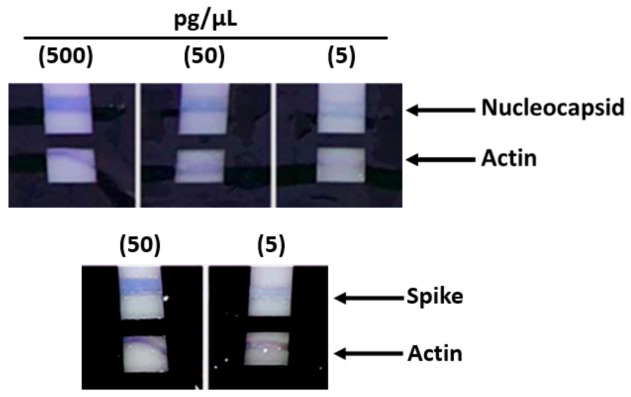
ELISA assays. The support with the strips was immersed in a solution containing a cell lysate (range: 500 pg/µL down to 5 pg/µL). These tests that were carried out with lower protein concentrations still showed a positive result, although with a pale pink color.

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
