# Peer review of "Detection of SARS-COV-2 Proteins Using an ELISA Test"

_diagnostics, 2021, doi:10.3390/diagnostics11040698_

Round 1
Reviewer 1 Report
The aim is stated clear. The authors stated clearly what study found and how they did it. The title is informative and relevant. Appropriate and key studies are included.
The study methods are valid and reliable. There are enough details provided in order to replicate the study.
The data is presented in an appropriate way. The text in the results add to the data and it is not repetitive. The conclusions answer the aim of the study. The conclusions are supported by references and own results.
Specific comments on weaknesses of the article and what could be improved:
Major points
- Figures 1 and 2 are not representative at all. Please, consider using other figures or removing these
- What are the estimated sensitivity and specificity of the developed tests?
Minor points
- Please, state the limitations of the study
- Could you please discuss the clinical implications of the results
Author Response
Point-by-point answers
Reviewer 1:
The aim is stated clear. The authors stated clearly what study found and how they did it. The title is informative and relevant. Appropriate and key studies are included.
The study methods are valid and reliable. There are enough details provided in order to replicate the study.
The data is presented in an appropriate way. The text in the results add to the data and it is not repetitive. The conclusions answer the aim of the study. The conclusions are supported by references and own results.
Response: We thank the reviewer for the appreciation to our work.
Specific comments on weaknesses of the article and what could be improved:
Major points
- Figures 1 and 2 are not representative at all. Please, consider using other figures or removing these.
Response: We agree with the reviewer that Figures 1 and 2 are not representative and therefore we have removed them as suggested.
- What are the estimated sensitivity and specificity of the developed tests?
Response: We thank the reviewer for this good suggestion. We have included additional information regarding the estimated sensitivity and specificity of the test in the "Results" section of revised version of the manuscript. In detail, the content of the additional part is as follows:
“These antibodies guarantee specificity towards the proteins under examination, because does not cross-react with SARS-CoV or MERS-CoV Nucleocapsid and Spike proteins based on internal testing performed by the productor”.
“Subsequent tests performed with protein concentrations lower than 5 pg, demonstrated the ability of the method to detect concentrations up to 2 pg, but in the practical use of the device in order to have an adequately detectable signal, we preferred to indicate 5 pg as the sensitivity limit”.
Minor points
- Please, state the limitations of the study.
- Could you please discuss the clinical implications of the results.
Response: We thank the reviewer for pointing out this lack. For this purpose we have added, in "Conclusion" section the following part:
“Our device allows you to perform a test that directly evaluates the presence of the virus in the biological sample taken through its specific proteins (Nucleocapsid and Spike). It is quick, in fact, it takes about 30 min. It offers the advantage that the result of the antigen-antibody reaction can be directly visible to the naked eye without the need to be carried out in a laboratory, without analytical stumentations and above all without the need to transport the samples. The high sensitivity of the method guarantees an early diagnosis even in those subjects with borderline SARS-CoV-2 virus positivity. Its signal cut off point makes it unique with respect to other diagnostic systems, in fact our test has a particularly low sensitivity limit. Another strength of our test not present in other is that of simultaneously detecting two specific viral antigens which certainly increase their specificity. Furthermore, the use of a cell lysis buffer in the first step of our kit allows to obtain a greater viral load in solution which accentuates the sensitivity of our method. In addition, it is a non-invasive test for which there is greater acceptability by the subjects; the ease of sample collection and the simplicity of execution means less need for operator training. Finally, the low cost allows for greater repetition. A limit, however, is that a sample can be negative if the concentration of the antigens is lower than the detection limit of the test, as it can happen if the sample is taken too early with respect to the hypothetical moment of exposure. For this reason, the negativity of the sample in the face of strong suspicion of Covid-19 should be confirmed by molecular test. So, the response speed of our test, the high sensitivity make it possible to monitor the incidence and progress of the disease as well as evaluate its severity over time and, moreover, allow to operate a mass screening or to photograph and crystallize the epidemic state of community environments such as schools, airports, stations, ports etc. preventing clusters or outbreaks”.

Reviewer 2 Report
The manuscript entitled: “Detection of SARS-COV-2 virus proteins by ELISA test”, presents a new method for the determination of the SARS-COV-2 virus in the biological samples. The manuscript is well written but needs some corrections before being ready for publication.
In the part describing the PVDF strip preparation protocol a figure that show the preparation process will be beneficial for an easy understanding of the procedure.
-It will need to specify the limit of detection for the tests
- Also the strengths and limitations of this method should be added and the comparison with the others methods used on the market. The sensitivity of this method compared with the other available methods should be discussed
Author Response
The manuscript entitled: “Detection of SARS-COV-2 virus proteins by ELISA test”, presents a new method for the determination of the SARS-COV-2 virus in the biological samples. The manuscript is well written but needs some corrections before being ready for publication.
Response:
We thank the reviewer for the appreciation to our work.
In the part describing the PVDF strip preparation protocol a figure that show the preparation process will be beneficial for an easy understanding of the procedure.
Response:
We thank the reviewer for this suggestion. We have added, in the revised form of manuscript, a new figure (Figure 1) describing the preparation of the PVDF strip.
New Figure:
Figure 1. Stages of preparation of the PVDF strip.
- It will need to specify the limit of detection for the tests.
Response:
As already asked by the reviewer 1 we have included information regarding the limit of detection of the test in the "Results" section of revised version of the manuscript. In detail, the content of the additional part is as follows:
“Subsequent tests performed with protein concentrations lower than 5 pg, demonstrated the ability of the method to detect concentrations up to 2 pg, but in the practical use of the device in order to have an adequately detectable signal, we preferred to indicate 5 pg as the sensitivity limit”.
- Also the strengths and limitations of this method should be added and the comparison with the others methods used on the market. The sensitivity of this method compared with the other available methods should be discussed.
Response:
As already asked by the reviewer 1 we have added, in "Conclusion" section the following part:
“Our device allows you to perform a test that directly evaluates the presence of the virus in the biological sample taken through its specific proteins (Nucleocapsid and Spike). It is quick, in fact, it takes about 30 min. It offers the advantage that the result of the antigen-antibody reaction can be directly visible to the naked eye without the need to be carried out in a laboratory, without analytical stumentations and above all without the need to transport the samples. The high sensitivity of the method guarantees an early diagnosis even in those subjects with borderline SARS-CoV-2 virus positivity. Its signal cut off point makes it unique with respect to other diagnostic systems, in fact our test has a particularly low sensitivity limit. Another strength of our test not present in other is that of simultaneously detecting two specific viral antigens which certainly increase their specificity. Furthermore, the use of a cell lysis buffer in the first step of our kit allows to obtain a greater viral load in solution which accentuates the sensitivity of our method. In addition, it is a non-invasive test for which there is greater acceptability by the subjects; the ease of sample collection and the simplicity of execution means less need for operator training. Finally, the low cost allows for greater repetition. A limit, however, is that a sample can be negative if the concentration of the antigens is lower than the detection limit of the test, as it can happen if the sample is taken too early with respect to the hypothetical moment of exposure. For this reason, the negativity of the sample in the face of strong suspicion of Covid-19 should be confirmed by molecular test. So, the response speed of our test, the high sensitivity make it possible to monitor the incidence and progress of the disease as well as evaluate its severity over time and, moreover, allow to operate a mass screening or to photograph and crystallize the epidemic state of community environments such as schools, airports, stations, ports etc. preventing clusters or outbreaks”.

Round 2
Reviewer 1 Report
The authors have addressed all the issues and the paper is improved significantly.
Reviewer 2 Report
The authors addressed all the comments received and now the manuscript is much improved compared with the initial version and is ready for acceptance.